# Red Light Enhances Plant Adaptation to Spaceflight and Mars *g*-Levels

**DOI:** 10.3390/life12101484

**Published:** 2022-09-24

**Authors:** Francisco-Javier Medina, Aránzazu Manzano, Raúl Herranz, John Z. Kiss

**Affiliations:** 1Centro de Investigaciones Biológicas Margarita Salas—CSIC, 28040 Madrid, Spain; 2Department of Biology, University of North Carolina at Greensboro, Greensboro, NC 27402-6170, USA

**Keywords:** light, microgravity, plant growth, root meristem, spaceflight, transcriptomics

## Abstract

Understanding how plants respond and adapt to extraterrestrial conditions is essential for space exploration initiatives. Deleterious effects of the space environment on plant development have been reported, such as the unbalance of cell growth and proliferation in the root meristem, or gene expression reprogramming. However, plants are capable of surviving and completing the seed-to-seed life cycle under microgravity. A key research challenge is to identify environmental cues, such as light, which could compensate the negative effects of microgravity. Understanding the crosstalk between light and gravity sensing in space was the major objective of the NASA-ESA Seedling Growth series of spaceflight experiments (2013–2018). Different *g*-levels were used, with special attention to micro-*g*, Mars-*g*, and Earth-*g*. In spaceflight seedlings illuminated for 4 days with a white light photoperiod and then photostimulated with red light for 2 days, transcriptomic studies showed, first, that red light partially reverted the gene reprogramming induced by microgravity, and that the combination of microgravity and photoactivation was not recognized by seedlings as stressful. Two mutant lines of the nucleolar protein nucleolin exhibited differential requirements in response to red light photoactivation. This observation opens the way to directed-mutagenesis strategies in crop design to be used in space colonization. Further transcriptomic studies at different *g*-levels showed elevated plastid and mitochondrial genome expression in microgravity, associated with disturbed nucleus–organelle communication, and the upregulation of genes encoding auxin and cytokinin hormonal pathways. At the Mars *g*-level, genes of hormone pathways related to stress response were activated, together with some transcription factors specifically related to acclimation, suggesting that seedlings grown in partial-*g* are able to acclimate by modulating genome expression in routes related to space-environment-associated stress.

## 1. Introduction

We have recently commemorated the 50th anniversary of the first time that a human being stepped on the surface of the Moon, the first terrestrial body outside the Earth in receiving humans. Nowadays, fifty years after Astronaut Armstrong’s epic achievement, space exploration by humans is commonly recognized as an exciting and attractive challenge and a powerful booster for scientific and technological progress in order to improve human life on Earth [1]. This is true despite some minor criticisms that question the high costs that it entails [2].

The establishment of permanent settlements in the Moon and Mars is becoming a realistic possibility in the near future. After a decade of successful rover explorations to the surface of Mars [3], both ESA and NASA, and more recently, the agencies from growing economies in Asian countries, are working to promote a human mission, first to the Moon, and then to Mars.

The objectives of deep space exploration by humans, including of the Moon and Mars, require the implementation of a complex system of life support for space explorers, capable of supplying the elements necessary for sustaining their life (oxygen, food, moisture, etc.) and of removing their waste products. The system needs to be bioregenerative, i.e., the components need to self-regenerate without the addition of new elements brought from Earth, and energy-efficient, only using the power sources available in space.

Plants are a candidate to occupy a key position in these bioregenerative life support systems (BLSSs). They are indeed being used in all the initiatives tested currently, such as MELiSSA—Micro-Ecological Life Support System Alternative [4,5]. There is no doubt that plants must accompany humans in space exploration ventures, because they offer the potential to provide food, replenish the air, filter water, and improve the psychological wellbeing of the crew during long-duration missions in space.

The achievement of a true “space agriculture” is a fundamental objective of this global enterprise, and the advances in this task are already producing substantial benefits for the efficiency and sustainability of the terrestrial agriculture [6,7]. The existence of “Martian greenhouses” is an image that appears to be more common to the public eye. In these greenhouses, plants could be provided with the necessary environmental elements to enable their development. These elements include light, water, temperature, oxygen, CO_2_, and aeration, in addition, obviously, to a substrate in which to anchor, from which the plant will take nutrients through its roots. Apart from anchoring the root, the substrate (e.g., soil) has to be capable of sustaining root development. Some published experiments have demonstrated that terrestrial plants can grow in an analogue of lunar soil or Martian soil, provided that this substrate is supplemented with additional elements and substances that provide the plant with the water, mineral salts, and nutrients that it needs for its survival and development [8]. Recently, plants have been grown in real Lunar regoliths brought back by Apollo missions, showing that the Lunar regolith is capable of supporting plant growth; however, it is not certainly the best substrate, because some stress was generated in samples grown on it [9].

In addition to the issues originated by the alien soils and the supply of the environmental conditions required for the growth of terrestrial plants, two important adverse factors need to be counteracted. These are the gravity level, different from the Earth value (microgravity—near 0 *g*—in orbit, 0.17 *g* for the Moon surface, and 0.38 *g* for the Mars surface) and the existence of cosmic radiation. These two are the only environmental factors that are unavoidable and unable to be counteracted using technological solutions or elements available on Earth. Moreover, plants have not encountered these conditions throughout their evolutionary history. Thus, the cultivation of plants in spaceflight, as well as on the Moon and Mars, requires that the specimens to be cultivated become adapted to grow and develop in the presence of these two environmental factors. To achieve this objective, we first need a full understanding of the biological mechanisms of response and adaptation to conditions of the spaceflight environment. Obviously, the development of technological countermeasures or the use of biological strategies that would mitigate the unfavorable impact of gravity and radiation is a complementary strategy to consider.

## 2. Altered Gravity/Microgravity Disturbs Plant Growth

Plant seeds were, strictly speaking, the first organisms in space, launched on a U.S. V-2 rocket in 1946, representing some early suborbital biological experiments handled by Harvard University and the Naval Research Laboratory [10]. Later, in 1966, the Soviet Kosmos 110 was launched, carrying two dogs and moisturized seeds; some of the seeds actually germinated. However, the first scientific experiment consisting of growing plants in space was carried out within the Oasis 1 hardware aboard the spacecraft Salyut 1 in 1971 [11], (reviewed by [12]). Then, experiments were performed on facilities installed in the NASA Space Shuttle, or on the Soviet MIR Station. As for space research in general, investigations on the response of plants to the space environment were significantly boosted after the assembly and operation of the International Space Station (ISS) [13]. In parallel, similar plant growth experiments on Earth, using ground-based facilities for microgravity simulation, such as clinostats and random positioning machines (RPMs), have successfully been performed [14,15]. In general, depending on the parameters studied, these have provided a reliable replica of the biological results obtained in space experiments under real microgravity, even though the gravity vector cannot possibly be avoided or removed on the Earth’s surface [15,16,17]. Only drop-towers provide a real microgravity on Earth, but the period of time for which the microgravity condition is generated in them is limited to just a few seconds [18], which is too short for plant research. Parabolic flights also provide a real microgravity for experimentation, but it is preceded by a period of hypergravity [19], which may alter the way an organism responds to microgravity. Suborbital flights, especially sounding rockets, have successfully been used for exposing plants to microgravity for limited periods of time (i.e., in the order of minutes) [13,20].

The main conclusion obtained from pioneering space experiments in orbit was that plants could survive and grow in space, although alterations were soon reported [21,22]. The results were sometimes contradictory, in most cases due to deficiencies in the experimental setup and the devices used to germinate seeds and grow plants. Major improvements in culture facilities have allowed us to conclude that microgravity itself does not prevent plant growth and reproduction. In fact, the use of newly implemented research facilities on the ISS led to impressive progress in plant biology research in space over the last two decades [13,23]. However, the number of pending unresolved questions is enormous and the difficulties and constraints of spaceflight research still represent major obstacles.

The first European experiment on plant biology, carried out on the ISS in 2003, was the ROOT experiment, included in the experimental program of the “Cervantes” Mission (Spanish Soyuz Mission) with the objective of investigating alterations induced by the space environment on cellular features of the root meristem. This experiment was performed using autonomous hardware based on the former Biorack containers [24,25], still seriously constrained regarding the environmental control of plant growth parameters [26]. Subsequently, a key step in this research on the ISS was the installation of the European Modular Cultivation System (EMCS), a facility specifically dedicated to the growth of seedlings and small plants equipped with advanced environmental controls for plant growth, as well as centrifuges for setting specific gravity conditions [27].

The first experiment carried out in the EMCS was the TROPI project, in 2006, aiming to study the interacting effects of gravity and other stimuli, such as light, on plant growth, to gain insights into the cellular and molecular mechanisms of tropisms and to better understand the interaction between gravitropism and phototropism [28]. Since then, the influence of gravity on early plant development and growth, as well as gravity perception thresholds for species such as lentils and *Arabidopsis*, and, more recently, the changes in the patterns of gene expression induced by the microgravity environment, have been studied on the ISS using the ESA’s EMCS and KUBIK incubators [29,30], NASA’s BRIC—Biological Research in Canisters [31], ABRS—Advanced Biological Research System [32] and APEX—Advanced Plant Experiment [33], and the JAXA’s PEU—Plant Experimental Unit [34].

All these studies have demonstrated serious alterations in the physiology and development of seedlings and young plants grown in space. The results of the ROOT experiment revealed that one of the most relevant effects of altered gravity is the disruption of the meristematic competence in cells of the root apical meristem [26,35]. This observation means that, under microgravity conditions, cell proliferation and cell growth appear uncoupled, losing their coordinated progress, which is characteristic of these cells under normal ground gravity conditions [36]. The cell proliferation rate is increased and cell growth, estimated by the rate of production of ribosomes, the cellular factories of protein biosynthesis, is depleted. Further complementary experiments performed on ground-based facilities for microgravity simulation, including sequential sampling at different growth times and the analysis of expression of some key genes of cell cycle regulation, have confirmed the uncoupling of cell proliferation and ribosome biogenesis caused by altered gravity, showing that microgravity induces serious alterations in the root meristem. This tissue plays a central role in establishing developmental patterns of the plant; therefore, the alterations induced by microgravity represent a serious compromise for plant development and, maybe, even for plant viability. Interestingly, the effects of gravitational stress are detected from the very beginning of germination, in two-day-old seedlings [37,38].

The observations and interpretations made on root meristems from seedlings have been confirmed and demonstrated in studies performed on cells cultured in vitro and grown in simulated microgravity. Flow cytometry analysis provided evidence for an accelerated cell cycle in cells grown in the random positioning machine (RPM). The final acceleration was the result of a shorter G2 phase and a slightly longer G1 phase. Analysis of gene expression showed a general downregulation of genes involved in the G2/M transition checkpoint and the upregulation of many genes controlling the G1/S transition. This is accompanied by the downregulation of significant genes regulating ribosome biogenesis and by the depletion of levels of nucleolin and fibrillarin, two nucleolar proteins considered markers of this process. In addition, a general depletion of the nuclear transcription was detected, accompanied by an increase in chromatin condensation, which was associated with changes in enzymes involved in the epigenetic regulation of gene expression [39,40,41]. According to data from many studies, the factor triggering the cascade of functional events that eventually resulted in the alteration of meristematic cell proliferation and growth and in the disruption of meristematic competence is the phytohormone auxin [42]. This hormone is a chief controller of the balance between cell proliferation and cell differentiation existing in meristems, which is the basis of the fundamental involvement of the meristematic tissue in plant development [43]. Moreover, auxin influences multiple aspects of plant growth and development, including the regulation of cell cycle progression and the coordination between cell growth and cell division [44]. From a more general perspective, auxin regulates the connection between stimuli perceived by the plant and the cellular responses to them [45].

The TROPI-1 experiment, the first experiment carried out in the EMCS, produced high-quality video downlinks of growing seedlings. It was shown that emissions with blue light might give an increased phototropic response in the hypocotyls. In addition, a novel positive phototropic response was observed after exposure to red light in hypocotyls and roots developed in microgravity [46]. In a complementary experiment with shared objectives, called TROPI-2, the positive phototropic curvature in hypocotyls and roots mediated by phytochrome, observed as a response to red light, was confirmed. At gravitational accelerations ranging from 0.1 *g* to 0.3 *g*, an attenuation of red-light-based phototropism of both roots and hypocotyls was observed [47]. Furthermore, from the frozen TROPI-2 samples returned, high-quality RNA was isolated and transcriptomic analysis was performed. Differences in expression between spaceflight samples and ground controls mostly affected genes involved in regulating cell polarity, cell wall development, oxygen status, and cell defense or stress. These differences represent the adaptive mechanisms of plants to the spaceflight environment [48].

The transcriptomic study carried out in the TROPI-2 project was part of a considerable effort made by different research groups directed to determine the effects of spaceflight on the plant genome, especially how microgravity conditions change gene and protein expression, using advanced -omics methodologies. Various experimental approaches have been assayed. Transcriptomic studies have used seedlings, plant organs, whole plants, and cultured cells of different types (established lines and callus cultures) exposed to real microgravity (spaceflight) or to simulated microgravity in ground-based facilities (see, for example, [33,49,50,51,52,53]). Collectively, these and other studies have demonstrated the complex responses in plants, involving reprogramming of the gene expression patterns. To date, specific genes of responses to gravity alterations have not been found; instead, genes known to participate in general mechanisms of the response to abiotic stresses have been shown to modify their expression in the microgravity environment. Genes coding for heat-shock-related elements, cell wall remodeling factors, oxidative burst intermediates, and components of the general mechanisms of plant defense are the main and most frequent targets of gene reprogramming induced by microgravity.

## 3. However, Plants May Overcome Gravitational Stress and Adapt to the Space Environment

All the aforementioned research has focused on studies of the alterations of biological mechanisms in plants induced by the space environment. Using molecular, cellular, and physiological methods, this has led to the conclusion that the space environment is deleterious for normal plant life, such as it is known on Earth. However, together with this fundamental research, mainly using model plant species such as *Arabidopsis thaliana*, an alternative approach has consisted of the direct in situ production of vegetable crops on the ISS, and their eventual use as fresh food to supplement the packaged diet of astronauts. This has been (and still it is) a specific objective of NASA. Obviously, food crops grown in space experience different environmental conditions (e.g., reduced gravity and elevated radiation levels) compared with plants grown on Earth, which have been revealed as potentially harmful to plant life. However, the first result of this novel approach was that red romaine lettuce was successfully grown in three tests in the Vegetable Production System (Veggie) facility with two different harvest methods, and yields were comparable to growth on Earth. The production and tasting of the first “space salad” and, shortly after, the development of a Zinnia flower, received much attention from mass media all over the world, and the first nutritional analysis of space-grown plants has recently been published [54], showing that lettuces grown on the ISS are nutritionally and microbiologically safe for humans. In addition to Veggie, NASA has recently implemented the Advanced Plant Habitat (APH) on the ISS, a more sophisticated facility with similar purposes, and complemented the Veggie facility with the Exposed Roots On-Orbit Test System (X-ROOTS). The general purpose of this effort is to perform compatible plant-related scientific research on ISS, using molecular, cellular, and physiological methods, with the direct production of fresh and nutritious food in bioregenerative life support systems (BLSSs). The final objective is that reduced plant growth in space can be directly translated into larger planting areas.

In addition to applications for the production of food for space explorers, these experiments enable the drawing of relevant conclusions related to the physiology of plants in microgravity. Lettuce and other crops cultured in Veggie and APH are indeed adult plants; therefore, despite the stresses on plant growth and development caused by the space environment, and specifically, by microgravity, plants are capable of overcoming adverse circumstances, removing the obstacles, and achieving successful development until the adult stage, including reproduction.

These results complement experiments specifically addressed to complete the full life cycle of plants (seed-to-seed) in space, carried out with *A. thaliana*. Using advanced growth chambers which, in general, provide a well-regulated environment for growing plants in microgravity on the ISS, fertile adult plants have been produced from seeds germinated in space; seeds obtained from these plants have, in turn, been germinated. The ADVANCED ASTROCULTURE (ADVASC) experiment, consisting of two successive experiments, was carried out in 2001–2002, in which the plant life cycle was completed [55]. A second experiment was the Japanese SPACE SEED, carried out in the Kibo module of the ISS in 2009 [34]. It is therefore clear that plants are capable of acclimating/adapting to the space environment. However, we currently lack knowledge of the mechanisms (cellular or molecular) by which the adaptation takes place and eventually succeeds. Currently, a key challenge for plant space research is to determine how and when the plant triggers mechanisms of adaptation in order to attenuate, or even overcome, the survival problems associated with a weightless environment.

## 4. Plant Culture in Space May Benefit from Other Environmental Cues Replacing Gravity

The optimization of plant growth in the microgravity environment of spaceflight, as well as in any other condition of reduced gravity, could become feasible by implementing the substitution of gravity by another external cue, which could play the same or a similar role in driving plant growth and development as gravity does on Earth. Light is a good candidate to be one of these cues because it is a tropistic stimulus. Phototropism complements gravitropism under normal ground conditions, with the objective of optimizing the efficiency of the capture of nutrients. In addition, illumination, especially by red light, is sensed and mediated by phytochromes to produce changes in the regulation of auxin-responsive genes and many growth coordinators [56]. A specific effect of red light in the activation of specific cellular processes known to be depleted in the microgravity environment, such as cell proliferation and ribosome biogenesis, had previously been reported [57]. Furthermore, light is known to induce the reorganization of chromatin architecture, as well as a global amplification of nuclear transcription [58]. These effects could potentially reverse the epigenetic de-activation that was found to be an effect of the simulated microgravity in cultured cells, as mentioned above.

Interestingly, the enhancement of light signaling to compensate for the absence of gravity was found to be a spontaneous response of plants to the lack of gravitropic stimuli in spaceflight, as shown by the upregulation of genes associated with plastid functioning, some of them closely related to photosynthesis [52]. This genetic response took place even in etiolated seedlings, in the absence of an effective light signal. The culture of seedlings in simulated microgravity under a photoperiod regime was capable of reverting many of the alterations found on etiolated seedlings incubated in the same facility for microgravity simulation [59]. Actually, light induces variable tropism in the aerial part of the plant (shoot and leaves; hypocotyl in seedlings) and in roots. Recent experiments have shown that seedling orientation in simulated microgravity can be improved by illuminating hypocotyls while keeping roots in darkness, by the action of a “light avoidance” mechanism that operates in roots [60]. This behavior, which mimics the natural situation of growing plants in soil, can be used advantageously to improve plant culture in space.

## 5. The Advantages of Red Light Illumination to Improve Seedling Viability in Space, as Revealed by the SEEDLING GROWTH Experiments

The series of three experiments, termed the SEEDLING GROWTH (SG) Project, was conducted on the ISS (2013–2017) with the aim of ascertaining the link between light and gravity, and of determining the combined influence of light and gravity on plant development, at reduced gravity levels. Among other objectives, the project investigated to what extent light could act as a signal capable of counteracting the deleterious effects caused by the lack of gravity. Specifically, we aimed to explore if, and to what extent, specific light conditions could be applied to modulate the alterations caused by the lack of gravity on plant growth and development, thus facilitating the adaptation of plants to the space environment.

The project consisted of three phases, termed SG1, SG2, and SG3, each performed in different spaceflights to the ISS, or ISS increments, between 2013 and 2017. The last experiment was the Ground Reference Control, carried out in 2018. This project was the result of the cooperation of NASA and ESA, using a European incubator (European Modular Cultivation System, EMCS) [27] equipped with two centrifuges, enabling control of the gravity level. The use of this spaceflight incubator was combined with culture chambers for the incubation of seeds and the growth of seedlings developed by NASA, and termed “TROPI cassettes” [28], and with a novel device, called FixBox, specifically designed and built in Europe for the chemical fixation of samples within cassettes to allow their microscopical examination postflight [61]. Different collections of mutants of *Arabidopsis thaliana*, affecting phytochromes, nucleolar proteins, and auxin-responsive genes were used, as well as the DII-Venus reporter line for the visualization of auxin localization [62]. Depending on the specific mutants, the corresponding wild-type ecotypes were Landsberg erecta (Ler), for the phytochrome mutants, and Columbia (Col-0) for all the other mutants and lines. Seeds germinated in-flight and grew for six days under different regimens of illumination and gravity. In addition to microgravity existing in space, seedlings were subjected to different levels of gravity between microgravity and 1 *g*, including Moon and Mars gravity levels, achieved with the on-board EMCS centrifuges.

In the SG1 and SG2 experiments, phytochrome mutants (*phyA* and *phyB*) and Ler wild-type seeds were germinated after hydration. Seedlings were initially grown at 1 *g* under white light for 96 h with the purpose of establishing a robust growth axis, followed by unidirectional (lateral) photostimulation for 48 h with red light, blue light, or red followed by blue light, under six gravity conditions: microgravity, 0.1 *g*, 0.3 *g*, 0.5 *g*, 0.8 *g*, and 1.0 *g*. Videorecording during the photostimulation phase enabled the identification of new phototropic responses to blue light in space, which complement findings obtained in previous TROPI I and II experiments [63].

Blue light is a known source of phototropic stimulus on Earth [64]. In plants grown in space under phototropic stimulation with blue light and under different gravity levels, which were preserved by ultra-freezing (−80 °C) and returned to Earth, a global transcriptomic study provided a very clear differential transcriptional response to each gravity level, from microgravity to 1 *g*. In the case of the microgravity-exposed plants, functions associated with light sensing and response, such as photosynthesis and related factors, appeared to be downregulated with respect to 1 *g* controls, suggesting that the growth was not following the phototropic environmental cue in the absence of the gravitropic cue. This observation would mean that gravity responses influence plant development under exposure to directional blue light. [65]. A similar analysis performed under the different levels of gravity mentioned above, including those corresponding to the Moon and Mars, showed that the effects induced by microgravity, especially those affecting plastid-related genes, were gradually removed by increasing the *g*-load, and that different functions appeared to be affected at different *g*-levels. However, a strong general abiotic stress response was detected at levels lower than the Moon gravity (lower than 0.1 *g*), probably due to the confluence of different altered stimuli at the detection threshold of photo- and gravi-sensing mechanisms, which could cause conflicting responses. At higher *g*-levels, the alteration became progressively weaker [66]. In general, blue light phototropism was found to be capable of reducing the gravitational stress response in spaceflight.

The effect of red light on Ler wild-type samples grown with the same protocol described above was investigated in roots by selecting marker genes for cell growth and proliferation and analyzing their differential expression under different conditions of gravity and light using qPCR. The results showed a positive effect of photoactivation with this wavelength in counteracting the stress caused by spaceflight in the root meristem, which had previously been identified and described. In these conditions, marker genes for cell proliferation, cell growth, and auxin polar transport showed a concerted upregulation [67]. Therefore, unilateral illumination with red light during the last two days of culture in microgravity on the ISS, after four days of growth under white light at 1 *g*, was capable of reverting, totally or partially, the alterations caused by microgravity on the root meristem. This included re-establishing meristematic competence and auxin transport. A parallel study on the ground using simulated microgravity (RPM) confirmed the spaceflight data [67].

In additional experiments of the series (SG2 and SG3), Col-0 seedlings and mutants were grown for 4 days under illumination with a photoperiod (16 h white light/8 h dark) and were then photostimulated with directional red light for two days. Control samples were kept in darkness for this period. Seedlings grew either in microgravity, at 0.3 *g* (a gravity level near that of Mars which was obtained with the centrifuge installed in the EMCS), and at 1 *g* (flight control), also produced by the rotating centrifuge. The effects of light and gravity on the growth of seedlings were noted on images taken after the 4-day period of growth and after the 6th day, including red light photostimulation (Figure 1).

Regarding wild-type seedlings, illumination in the photoperiod showed a differential effect on hypocotyls and roots. Under microgravity, hypocotyls appeared generally oriented towards the light source, showing little differences with the orientation exhibited by control seedlings grown at 1 *g*. In contrast, light did not appear to influence the orientation of roots in microgravity, which showed considerable differences with the oriented roots of seedlings grown at 1 *g* (Figure 1). These features are in agreement with the results of experiments performed in simulated microgravity [59]. Photoactivation with directional red light suppressed the orientation of hypocotyls achieved under white light. In general, seedlings appeared randomly oriented, although the effect was stronger in seedlings located closer to the red light source, whereas seedlings placed further from light preserved some orientation (Figure 1). Red light phototropism, described in the roots of seedlings grown in microgravity [46], was weak; thus, it was difficult to observe when the starting point was disoriented roots. Nevertheless, signs of this root tropism could be detected in some seedlings (Figure 1). The images of seedlings grown at 0.3 *g* were especially significant, because they showed seedlings almost totally oriented in their hypocotyls and roots according to the partial gravity vector. This orientation, very similar to the result of growth at 1 *g*, was not suppressed by red light, indicating that it was a clear effect of this level of gravity (Figure 1). Thus, the gravity vector existing at the surface of Mars is capable, *per se*, of inducing a positive gravitropism in the roots of *A. thaliana.* This finding could be relevant to realizing the culture of plants on the Red Planet.

A full-genome global transcriptomic analysis of red-light-photostimulated plants *versus* plants grown in darkness was performed on samples grown in the SG2 spaceflight experiment. A specific purpose of this analysis was to find answers to two relevant questions: (a) Is red light irradiation capable of reverting the pattern of gene expression under microgravity to a pattern closer to that of control 1 *g* conditions?; and (b) Is red light irradiation capable of counteracting gravitational stress? Transcriptomic data were analyzed using a bioinformatics tool called Gene Expression Dynamic Inspector (GEDI v2.1) [68]. The GEDI profile enabled visualization of the gene expression across the transcriptome, generating a mosaic image or dot matrix, using a self-organizing map algorithm [68]. GEDI creates clusters of genes that share dynamic expression patterns, irrespective of gene ontology or cellular function. The output of GEDI is a characteristic mosaic or GEDI Map, a color-coded two-dimensional grid image (20 × 16 pixels) for each sample (Figure 2a). Each tile in the mosaic (each pixel) represents a cluster of genes that share a similar transcriptional profile in any experimental condition. The color of each tile indicates the relative expression of the genes in that cluster. Finally, clusters of similarly expressing genes are placed in the same neighborhood on the grid, creating an image that enables global transcriptome analysis as a single entity for display in different conditions.

Global patterns of the gene expression of seedlings grown for six days in the different conditions of the SG2 experiment, represented as GEDI maps, allowed the comparison of the effects of gravity and red light photoactivation (Figure 2a). The upper row shows the patterns obtained for the full genome (22,719 genes analyzed). The pattern of seedlings grown at 1 *g* under red light photoactivation was significantly changed in seedlings grown in microgravity and not photoactivated. Interestingly, the pattern was reversed under microgravity when samples were photoactivated, indicating a possible effect of red light in compensation of the alterations induced by microgravity (Figure 2a). Unfortunately, the samples in darkness and at 1*g* were lost during processing on the ISS and could not be incorporated in the analysis. If the same comparison was restricted to the set of genes showing differential expression in microgravity *versus* ground gravity (12,950 genes), differences in the pattern were increased and the reversion induced by red light in microgravity was only partial (Figure 2a).

Transcriptomic data were subjected to gene ontology (GO) analysis in order to identify the functions especially altered by the different conditions and to differentiate downregulated and upregulated genes (Figure 2b). The results of this analysis were compared, first under constant microgravity, between red light photoactivation and darkness, and then under constant red light photoactivation, between microgravity and ground gravity. Differences in gene expression, caused by either up- or downregulation, in the different ontology groups were much greater in the first case than in the second case (Figure 2b). The interpretation was that red light photoactivation attenuated the differences in gene expression induced by microgravity. Under constant microgravity, there was a profound difference in gene expression between photoactivated and non-photoactivated samples, but red light illumination significantly mitigated the differences caused by the gravity change (Figure 2b).

In these studies, we also used mutants of the nucleolar protein nucleolin, a multifunctional protein that plays key roles in the regulation of the fundamental cellular process of ribosome biogenesis, at multiple steps [69]. The genome of *Arabidopsis thaliana* encodes two genes for nucleolin, called *NUC1* and *NUC2.* From these two nucleolin genes*,* one of them, *NUC2*, is known to participate in the mechanisms of adaptive responses to different stresses [70]. We took advantage of this feature to evaluate the role of red light photoactivation in the attenuation or mitigation of stress caused by microgravity on seedlings. For this purpose, we analyzed differences in the patterns of gene expression, expressed as GEDI maps, between the Col-0 wild type and the *nuc2* mutant, under different conditions of gravity and light in spaceflight. The differences in response between the two genotypes should be attributed to the role played by the Nuc2 protein in the stress response, because normal expression of the *NUC1* gene should be sufficient to ensure normal regulation of ribosome biogenesis. Under 1 *g* ground gravity and red light photoactivation, the GEDI map revealed very minor differences in gene expression between the two compared genotypes (Figure 2c). These conditions were not identified as stressful; therefore, there was no need for triggering the mechanisms of stress response mediated by the Nuc2 protein. For this reason, the presence or absence of the *NUC2* gene was irrelevant for the global pattern of gene expression. In contrast, microgravity and no photoactivation by red light were identified as stress-generating conditions in cells, capable of triggering the mechanisms of stress response in which the Nuc2 protein is involved. Consequently, the presence or absence of the *NUC2* gene greatly affected this response. The result was a significant difference in the global pattern of gene expression, revealed by the GEDI map, between the Col-0 wild type, in which the *NUC2* gene is expressed, and the *nuc2* mutant, which does not express this gene (Figure 2c). These differences in gene expression were considerably attenuated under microgravity and red light photoactivation in spaceflight, although the GEDI map of the comparison was not totally reversed to the pattern indicative of a full absence of stress (Figure 2c). The conclusion was that red light photoactivation could significantly contribute to realizing the adaptation of plants to the microgravity environment; therefore, it could be used to improve the culture of plants in space.

These results were confirmed and extended by a complementary analysis of the nucleolin mutants *nuc1* and *nuc2*, together with the Col-0 wild type, which was performed using samples grown in the SG3 experiment. In parallel, the response of these genotypes to red light photoactivation was investigated under 1 *g* ground gravity [71]. The basis of these studies was the finding that plant nucleolin gene expression is upregulated by red light on Earth [57]. A global transcriptomic study performed on the ground using WT and both *nuc1* and *nuc2* mutant lines, comparing conditions of red light photoactivation for two days versus darkness, revealed that illumination at this wavelength enhanced the capability of the *NUC2* gene to replace the functions of *NUC1* in ribosome biogenesis and cell cycle, when this gene was knocked out in the *nuc1* mutant. In addition, in the *nuc2* mutant, photoactivation induced an increase in differentially expressed genes (DEGs) belonging to functional groups associated with stress responses. This means that the *NUC2* gene may counteract the environmental stress produced by darkness in *nuc1* seedlings, although *nuc2* seedlings cannot develop a full response to red light [71].

Furthermore, the analysis of these samples grown in spaceflight (SG3) comprised, first, a phenotypic analysis including morphometric parameters of seedlings during development in space in combination with microscopical studies of the root meristems on samples fixed in orbit [61]. Study at the cellular level of the cell cycle and ribosome biogenesis in root meristematic cells resulted in values closer to the 1 *g* control in samples stimulated with red light. Then, gene expression alterations were evaluated by RNAseq. The *nuc2* mutant showed differential requirements in response to red light photoactivation and exhibited normal development in space by changing the expression of a smaller number of genes than the wild type, i.e., with reduced requirements in the number of DEGs, which could mean a better capacity than the wild type for adaptation to a microgravity environment. Therefore, a mutant line with this attenuated response may constitute an advantage to be taken into account when selecting the most productive plant varieties for life support systems, and opens the way to directed-mutagenesis strategies in crop design to be used in space colonization scenarios (Manzano et al., 2022, submitted for publication).

In addition, the localization of nuclear proteins and auxin distribution were analyzed by confocal and electron microscopy, in studies that were also performed on SG3 samples fixed in space on the ISS. Confocal microscopic images of the root tips of seedlings of the DII-Venus reporter line of *A. thaliana* showed an altered pattern of auxin distribution in seedlings grown in microgravity with respect to the 1 *g* control, indicative of some inhibition of the auxin polar transport. This finding confirmed previous experiments on simulated microgravity [72]. In contrast, the pattern of roots grown at 0.3 *g* was basically similar to the control pattern, indicating that auxin polar transport was not altered at these partial-*g* levels [73]. This is in agreement with the images of seedlings grown for six days on the ISS under this level of gravity (near-Mars *g*), showing that roots and hypocotyls appeared to be oriented following the direction of the gravity vector (Figure 1) and, consequently, that normal gravitropism was operating at these levels of gravity.

Samples of the Col-0 wild-type ecotype, grown in the SG3 experiment, were also analyzed by confocal and electron microscopy, together with transcriptomic analysis, to obtain additional data on the response of seedlings to microgravity and Mars gravity levels and on the effects of red light photostimulation at these two gravity levels, compared with 1 *g* ground gravity [74]. The cellular analysis was focused on root meristematic cells. Red light increased meristem and nucleolar size, as observed in the confocal microscopy images. In addition, ultrastructural analysis of meristematic cell nucleoli showed more active nucleoli in the red-light-photostimulated seedlings, as shown by the relative distribution of the nucleolar subcomponents, according to previously described models of the nucleolar structure in relation to activity [75,76]. This observation suggests that red light indeed counteracts the decoupling between cell proliferation and growth, as reported in the aforementioned prior experiments in space and simulated microgravity [35,37].

The transcriptomic RNA-seq analysis, globally considered, showed, as expected, that microgravity has a considerable impact on the transcriptome in *A. thaliana* seedlings. However, interestingly, partial gravity influences the transcriptome differently from microgravity. Red light photostimulation modulates the response to microgravity and also to partial gravity, producing a clear attenuation of the functional changes induced by altered gravity. Specifically, the transcriptomic analysis of phytohormone signaling indicated an activation of proliferation-promoting pathways in microgravity, such as cytokinin and auxin [74]. Cytokinin alterations in spaceflight were reported previously [77], and the changes in auxin polar transport in the different conditions of light and gravity, as detected in situ with the reporter line DII-Venus, are discussed above.

An important result of this analysis was observation of the downregulation of photosynthetic functions in red-light-photostimulated seedlings in microgravity, but not at the Mars gravity level [74]. This result was also obtained from seedlings grown in microgravity and photostimulated with blue light in the course of the SG1 experiment [65]. In both cases, the GO analysis exhibited downregulation in PSI and PSII, the photosynthetic electron transport chain, and chlorophyll metabolism. In addition, overexpression of the plastid and the mitochondrial genome was observed in microgravity.

Overexpression of the plastid genes had previously been interpreted as an attempt by the seedlings to enhance light signaling to compensate for the lack of gravity in dark-grown seedlings [52]. However, this overexpression was recorded at different light conditions, and it was indeed more pronounced in red-light-photostimulated seedlings than in seedlings that were kept in the dark. This observation indicated that this effect was not exclusive for etiolated seedlings [74]. The downregulation of photosynthesis functions in seedlings with red or blue light photostimulation could, in fact, be a result of altered plastid function due to alterations in nuclear–organelle communication. These alterations in nuclear–organelle communication could account for the elevated plastid and mitochondrial genome expression, which was only observed in microgravity conditions, both in darkness and under red light photoactivation. There are two types of nucleus–organelle communication, namely, anterograde (nucleus-to-organelle signaling) and retrograde (organelle-to-nucleus signaling). The dysregulation could take place in either of them: anterograde if the nucleus response does not reach the organelles; retrograde if the nucleus does not perceive the upregulation of plastid and mitochondrial genome expression [78].

However, the Mars gravity level corrected these effects and promoted an adaptive response, with the upregulation of genes of hormone pathways associated with the stress response (abscisic acid, ethylene, and salicylic acid). Furthermore, the upregulation of environmental-acclimation-related transcription factors (WRKY and NACs families) [79,80] was also observed, especially in samples photostimulated with red light, suggesting that seedlings grown in partial gravity levels are able to acclimate by modulating genome expression in routes activated to cope with space-environment-associated stress [74]. Strikingly, samples grown at near-Mars g (0.3 *g*) showed significant alterations in gene expression, more than expected in view of the data obtained in the SG1 experiment, as noted above [65,66]. There are differences between the two sets of samples (SG1 and SG3), which could be due to the different results, namely, ecotype, light conditions, and time of exposure to altered gravity, among others. Despite the differences, the Mars gravity level was found to cause a more adaptive response than microgravity on SG3 samples. In addition, the upregulation of WRKY transcription factors at the Mars gravity level showed that adaptive mechanisms were activated [74].

Collectively considered, the results showed a differential response to each gravity level, triggering different adaptive responses, involving changes in the regulation of different sets of genes. These changes in gene expression were more reduced under Mars gravity than under microgravity conditions. In all cases, the adaptive response appeared to be enhanced by red light photostimulation.

## 6. Conclusions and Future Prospects

The SG series of experiments has constituted a significant contribution to furthering the knowledge of alterations induced by the space environment, specifically by microgravity and different levels of partial gravity, on the physiology of plants. Investigations have focused on the earliest stages of plant development, and a variety of cellular and molecular (transcriptomic) parameters have been studied. Among the key findings of our spaceflight project are the new results on phototropism induced by red and blue light illumination, the effect of red light illumination in counteracting the adverse effects of altered gravity, and the discovery that Moon gravity could trigger more severe transcriptomic responses than spaceflight. However, we also observed that Mars gravity is capable of inducing the gravitropic orientation of roots, and the different responses to the space environment of the various mutants used, suggesting that the selection of plants for their culture in alien environments should require prior detailed analysis of their genotypes in order to select the varieties capable of adapting better.

The SG experiments have been among the first in performing a comparative analysis, using microgravity, different levels of partial gravity (including Moon and Mars gravity), and an on-board 1 *g* control in flight. This has been possible due to the use of the EMCS incubator, equipped with centrifuges providing this capacity. Unfortunately, the EMCS was decommissioned in 2018 and not replaced by any other spaceflight facility with these exact capabilities. The novel advanced NASA incubators for plant culture, Veggie, APH, and X-ROOTS, do not allow for these comparative studies at different gravity levels, which are of the highest importance.

In the immediate future, in our next studies, we should incorporate other space environmental factor that, along with gravity, is key to the survival of living beings and particularly of plants: radiation. Fortunately, there are experiments under development in different laboratories, including ours, and different devices and facilities providing the simultaneous application of radiation and different levels of simulated microgravity and partial gravity for experimental analyses.

In addition, we need to extend the cellular and molecular studies on plant adaptation to the space environment, such as SG, to the use of crop species. NASA is actively promoting plant studies on the ISS using the aforementioned spaceflight incubators Veggie, APH, and X-ROOTS, but this activity is basically oriented to nutritional aspects, without giving the necessary priority to fundamental studies at cellular and molecular level. Surprisingly, the activity of ESA in plant biology studies has been relatively slow after completion of the SG project. Obviously, the decommission of EMCS, an ESA facility, has been an important factor, but not the only one. There is an ongoing project to adapt the Biolab, a seriously underused spaceflight facility, to plant experiments of this type. It is vital that Europe regains the initiative in this research area as soon as possible, and thus again being able to play a relevant role in meeting the need for an effective culture of plants in space. Nowadays, this need, as a key human life support mechanism in space exploration, is becoming more and more urgent.

## Figures and Tables

**Figure 1 life-12-01484-f001:**
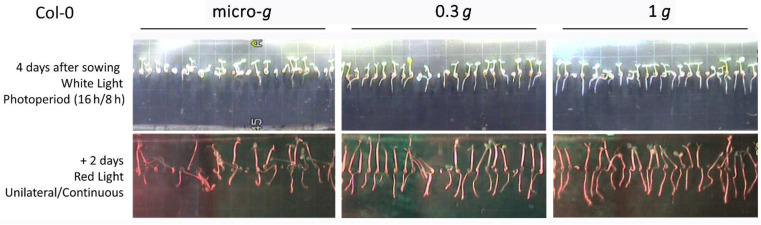
Images of *Arabidopsis thaliana* wild-type (Col-0) seedlings grown on the ISS in the Seedling Growth-3 (SG3) experiment. Seeds were affixed to a black gridded membrane (the distance between the grid lines on the membrane was 3 mm) and hydrated to promote germination. Upper row: seedlings grown for 4 days under a white light photoperiod (16 h/8 h). The light source was positioned in the upper part of the images. Different levels of gravity were applied using the centrifuges installed in the EMCS facility, namely, microgravity (micro-*g*), near-Mars gravity (0.3 *g*), and Earth gravity (1 *g*). Notably, in all cases, white light was capable of determining the orientation of hypocotyls; however, the orientation of roots depended on the level of gravity. Lower row: seedlings grown for two additional days under continuous unilateral illumination with red light. The light source was positioned at the left side of the images. Red light was not capable, *per se*, of inducing orientation of seedlings, as shown in the microgravity image; only a few seedlings appeared to orient their root tips towards the light source. However, partial gravity, as well as Earth gravity levels, were sufficient to produce the seedling orientation according to the gravity vector.

**Figure 2 life-12-01484-f002:**
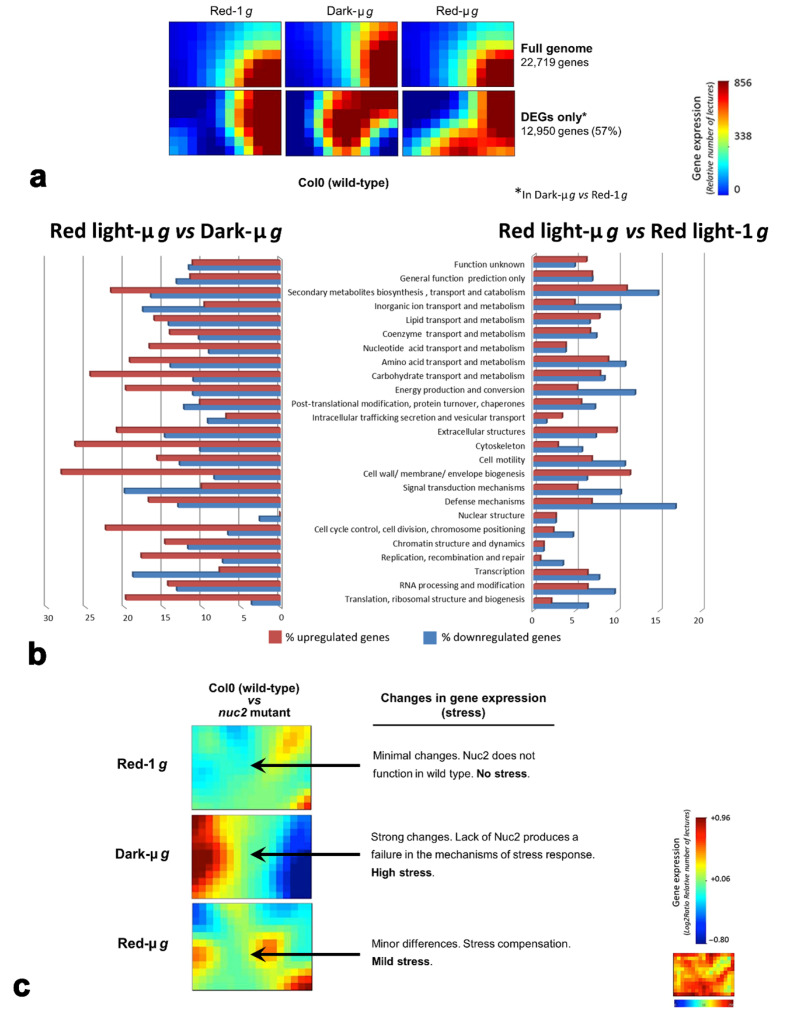
Analysis of the transcriptomic data obtained from samples grown in the Seedling Growth-2 (SG2) experiment on the ISS. Seedlings were grown for 4 days under a white light photoperiod (16 h/8 h), and were then either photoactivated with continuous unidirectional red light for an additional 2 days, or kept in darkness. (**a**) Global transcriptomic patterns obtained with the bioinformatic tool called Gene Expression Dynamic Inspector (GEDI) of Col-0 wild-type samples grown under different conditions. Genes showing similar degrees of expression were treated together, attributed a color code, and are presented as a tile or pixel in the diagram. Upper row: full genome. Lower row: differentially expressed genes (DEGs) in microgravity, with respect to 1 *g*. The pattern significantly changed from red-light-1 *g* conditions to dark-micro-*g* conditions, but it was reverted in micro-*g* with red light photoactivation. Reversion was higher for the full genome than for DEGs only. (**b**) Comparison of the transcriptomic effects of a change in illumination, at constant micro-*g* (left), and of a change in *g* at constant red light (right). Diagrams show the percentages of upregulated and downregulated genes in the different ontology groups. (**c**): GEDI maps, under different conditions of light and gravity, of the comparison in gene expression of the Col-0 wild-type and the *nuc2* mutant, defective in a gene participating in the stress response mechanisms. Differences in gene expression between the two genotypes in each condition were used to estimate the stress of this condition.

## Data Availability

Data are available upon reasonable request from the corresponding author.

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
