# Peer review of "Red Light Enhances Plant Adaptation to Spaceflight and Mars *g*-Levels"

_life, 2022, doi:10.3390/life12101484_

Round 1

Reviewer 1 Report

The present review by Medina et al. is an excellent attempt to provide the comprehensive overview of plant research going on space station and the effect of microgravity on gene expression and plant development. The manuscript is very well written, drafted and self-explanatory that attracts to general readers as well. It is interesting to know that the effect of red light illumination in counteracting the adverse effects of altered gravity. However, it would be exciting to understand whether this is the outcome of phytochromes or downstream signalling cascade as blue light also displayed photo-stimulation. I recommend this manuscript to be accepted. Few minor comments are:

  1. Line 107 and Line 230. So many dots. It need to be removed.
  2. The title of the manuscript should be precise and summarize the story. Authors need to rewrite the title.

Author Response

Reviewer 1:

  • Dots in lines 107 and 230 (headings of chapters 2 and 3 respectively) are not regular dots, but a rhetoric figure called “ellipsis”. In fact, the headings of chapters 2 and 3 form part of a single sentence: “Altered gravity/Microgravity disturbs plant growth, but plants may overcome gravitational stress and adapt to the space environment”. This single sentence has been split into two, which were separated by this “ellipsis” (“Altered gravity/Microgravity disturbs plant growth … but plants may overcome gravitational stress and adapt to the space environment”. Each term was explained separately in a different chapter.
  • The title of the paper has been shortened and made more concise by deleting the second part. Thus, our shortened title now is: “Red light enhances plant adaptation to spaceflight and Mars g-levels”

Reviewer 2 Report

In the review paper “Red light enhances plant adaptation to spaceflight and Mars g-2 levels. Results of the Seedling Growth experiments on the In-3 ternational Space Station”, authors reported a detailed list of works aimed at understanding how plants respond and adapt to extraterrestrial conditions. A full paragraph (Paragraph 5) has been dedicated to the description of a transcriptomic experiment that has been conducted to mine plant response to microgravity under different light wavelengths.

The whole work is very well described and worth to be published but, considering the quality of the results of the RNA-Seq experiment, the reviewer suggests preparing a conventional research paper for this part. Once the RNA-Seq paper will be published the author could resubmit a new review, including the former paper and considering all the issues that have been reported in the Conclusion paragraph in more details.

The reviewer is aware that this will require long time and intensive work, but he/she is sure that the data presented here by authors will result in a successful paper.

 Good work

Author Response

Reviewer 2:

First of all, we have to say that the Paragraph 5 (or Chapter 5) not only describes a transcriptomic experiment, as indicated by the reviewer, but it summarizes all the analyses, at the cellular and the molecular level, carried out on the samples recovered from the three experiments of the “Seedling Growth” project performed in the ISS. These included six transcriptomic studies (one by qPCR and five by RNAseq) as well as different cytological studies, using both light and electron microscopy, even with the use of reporter lines. All these analyses, except one, have been published in peer-reviewed journals and are conveniently cited in the article.

The suggestion of the reviewer is meaningful, and we are grateful to him/her for this. However, the proposed schedule would seriously alter the editorial planning designed for the present paper, i.e. to respond to the invitation sent by the guest editors to contribute to the special issue “Plants and Microgravity” of the journal “Life”.

Round 2

Reviewer 2 Report

The paper is accepted as it is